# Multi-View Clustering Based on Deep Non-Negative Tensor Factorization

## ABSTRACT

Multi-view clustering (MVC) methods based on non-negative matrix factorization (NMF) have gained popularity owing to their ability to provide interpretable clustering results. However, these NMF-based MVC methods generally process each view independently and thus ignore the potential relationship between views. Besides, they are limited in the ability to capture the nonlinear data structures. To overcome these weaknesses and inspired by deep learning, we propose a multi-view clustering method based on deep non-negative tensor factorization (MVC-DNTF). With deep tesnor factorization, our method can well exploit the spatial structure of the original data and is capable of extracting more deep and nonlinear features embedded in different views. To further extract the complementary information of different views, we adopt the weighted tensor Schatten $p$-norm regularization term. An optimization algorithm is developed to effectively solves the MVC-DNTF objective. Extensive experiments are performed to demonstrate the effectiveness and superiority of our method.

## CCS CONCEPTS

• **Computing methodologies** → **Machine learning**; *Artificial intelligence.*

## KEYWORDS

Multi-view clustering, Multi-view learning, Non-negative tensor factorization, Schatten $p$-norm

## 1 INTRODUCTION

In recent years, the advancement of information technology has led to an exponential increase in data generated across various domains. This data usually comes from different sources [1]. Multi-view data processing have demonstrated significant applications in scenarios such as image processing, social network analysis, and bioinformatics. As an unsupervised learning method, multi-view clustering gains increasing attention due to its ability to utilize the multiple data views to improve the clustering performance by exploiting the consistent and complementary information among them [15]. So far, numerous multi-view clustering methods are proposed, which can be roughly divided in four categories [41]: Co-training based methods [20], multi-kernel learning [34], multi-view graph clustering [39], and multi-view subspace clustering [45].

Permission to make digital or hard copies of all or part of this work for personal or classroom use is granted without fee provided that copies are not made or distributed for profit or commercial advantage and that copies bear this notice and the full citation on the first page. Copyrights for components of this work owned by others than the author(s) must be honored. Abstracting with credit is permitted. To copy otherwise, or republish, to post on servers or to redistribute to lists, requires prior specific permission and/or a fee. Request permissions from permissions@acm.org.

*ACM MM, 2024, Melbourne, Australia*

© 2024 Copyright held by the owner/author(s). Publication rights licensed to ACM.
ACM ISBN 978-x-xxxx-xxxx-x/YY/MM
https://doi.org/10.1145/nnnnnnn.nnnnnnn

Among these methods, the co-training approach [22] leverages the co-training framework to ensure consistency in either prior knowledge or clustering outcomes. Furthermore, Tzortzis et al [34] proposed a multi-view clustering technique based on kernel functions, which weights all views using kernel functions. This approach assigns different weights to various views, achieving improved clustering results. Pan et al [30] proposed a generalized framework for clustering multi-view attribute graph data and introduced the Multi-view Comparison Graph Clustering (MCGC) method, which enhances graph clustering by filtering noise and learning consensus graphs with contrast loss. Additionally, some approaches focus on individually optimizing each view to identify the most informative subspaces, enabling clustering within these subspaces. This strategy enhances the handling of high-dimensional data and noise, thereby improving the robustness and effectiveness of clustering algorithms [6, 35].

Notably, non-negative matrix factorization (NMF) has received widespread attention owing to its outstanding clustering performance and interpretability [8, 23], and many improved methods were developed based on NMF [3, 7, 9, 10, 37, 38], *etc.* For example, Cai et al [4] proposed a graph-regularized NMF (GNMF) method, which mines the flow structure of NMF data for effective clustering. Its effectiveness motivates researchers to extend NMF to multi-view NMF [46]. For example, Ou et al [29] introduced NMF into multi-view clustering by considering the geometric structure of each view to mine the potential information of each view; Semi-supervised multi-view clustering method based on NMF was proposed in [28], which adopts semi-supervised approach for clustering, and the effectiveness of the method is verified by several sets of experiments. Effective as they, these methods are incapable of extracting nonlinear features.

To overcome this weakness, Zhao *et al.* proposed a multi-view clustering via deep matrix factorization, which fully explores the potential attributes of the data in a layer-wise fashion and explore the geometrical structure with a graph-based regularizer [43]. Similarly, Zhang *et al.* developed a deep NMF based multi-view clustering method with partition alignment (MVC-DMF-PA), which employs deep NMF to learn the clustering partition matrix for each view and then fuse them into the final partition result with a partition aligment [42]. These deep NMF-based MVC methods exhibit impressive feature extraction ability. Besides, they realize better efficiency when compared with other MVC methods (*e.g.,* graph-based methods or deep methods). Nonetheless, they essentially apply NMF on each single view and then integrate them into a consistent clustering assignment matrix, resulting in the the loss of the original inter-view spatial information [24]. For another, since they directly decompose the original data, their efficiency decreases dramatically when the original data dimensions are large.

Kilmer *et al.* introduced the factorization of nonnegative tensors [21], which has been proven to be an effective to learn the

inter-view information from multi-view data and numerous tensorial MVC methods are developed [24, 27, 31, 44]. Inspired by their impressive performance, we in this paper extend deep NMF for multi-view clustering to deep non-negative tensor factorization (DNTF) and propose a novel multi-View clustering with deep tensor factorization (MVC-DTF). MVC-DTF not only takes into account the differences and complementarities between views but also enables the learning of deeper representations of the data through the deep tensor factorization method, so as to capture and utilize the complex structural information in the multi-view data in a more effective way. In addition to this, we use weighted tensor Schatten $p$-norm to describe the clustering structure of the view data, which fully utilizes the complementary information between views and avoids the problem that the use of Schatten $p$-norm in [24] leads to the same contribution of singular values across views. To further improve the efficiency, we perform tensor factorizaton on the anchor graph instead of the original data tensor. The main contributions of this paper are as follows:

- We extend deep non-negative matrix factorization to deep tensor factorization , based on which we propose a novel multi-view clustering method, called MVC-DTF. By further introducing a Schatten $p$-norm based regularizer, MVC-DTF can better exploit the inter-view spatial relationship and the complementary information of differentviews.
- We adopt anchor graph to construct the tensor to be factorized to reduce the computational complexity. Besides, we develop an optimization algorithm to solve the MVC-DTF optimization problem.
- Extensive experiments are conducted on several multi-view datasets to evaluate the prformance of our method, and the comparison with existing MVC methods demonstrate its effectiveness and superiority.

## 2 RELATED WORK

In this section, we introduce the concepts and formulas of multi-view clustering using non-negative matrix factorization and non-negative tensor factorization respectively, which is tightly related to our method.

### 2.1 Non-negative Matrix Factorization (NMF)

NMF has become an important feature dimensionality reduction tool in clustering. NMF provides an intuitive way to explore the low-dimensional structure of the data by decomposing the non-negative data matrix $\mathbf{X}$ into a weight matrix $\mathbf{W}$ and an indicator matrix $\mathbf{H}$. Its formula is expressed as follows:

$$\min_{\mathbf{H},\mathbf{W}} \|\mathbf{X} - \mathbf{H}\mathbf{W}\|_F^2, \text{ s.t. } \mathbf{W} \geq 0, \mathbf{H} \geq 0 \qquad (1)$$

where $\mathbf{X} \in \mathbb{R}^{n \times d}$, $\mathbf{H} \in \mathbb{R}^{n \times k}$ and $\mathbf{W} \in \mathbb{R}^{k \times d}$. Furthermore, Semi-NMF improvies NMF by relaxing the non-negative constraint of $\mathbf{W}$, which has been proven to be a relaxation of the K-means clustering approach [10]. Specifically, Semi-NMF can be expressed as follows:

$$\min_{\mathbf{H},\mathbf{W}} \|\mathbf{X} - \mathbf{H}\mathbf{W}\|_F^2, \text{ s.t. } \mathbf{H} \geq 0 \qquad (2)$$

## 2.2 Tensor Factorization

Kilmer *et al* [21] extended NMF to third-order tensors. Specifically, they present a new third-order tensors representation and introduce a closed tensor multiplication operation, based on which they defined tensor transpose, inverse, and unit tensor.

DEFINITION 1 ( [21]). *For a 3-way tensor $\mathcal{A} \in \mathbb{R}^{n_1 \times n_2 \times n_3}$, the Frobenius norm of $\mathcal{A}$ is defined as $\|\mathcal{A}\|_F = \sqrt{\sum_{ijk} |a_{ijk}|^2}$, and the conjugate transpose of $\mathcal{A} \in \mathbb{R}^{n_1 \times n_2 \times n_3}$ is $\mathcal{A}^T \in \mathbb{R}^{n_2 \times n_1 \times n_3}$.*

DEFINITION 2 ( [21]). *For a 3-way tensor $\mathcal{A} \in \mathbb{R}^{n_1 \times n_2 \times n_3}$, we denote $\overline{\mathbf{A}}$ as a block diagonal matrix with each block on the diagonal as the frontal slice $\overline{\mathbf{A}}^{(i)}$ of $\overline{\mathcal{A}}$. $\overline{\mathbf{A}}$ has the following form:*

$$\overline{\mathbf{A}} = \text{bdiag}(\overline{\mathcal{A}}) = \begin{bmatrix} \overline{\mathbf{A}}^{(1)} & & & \\ & \overline{\mathbf{A}}^{(2)} & & \\ & & \ddots & \\ & & & \overline{\mathbf{A}}^{(n_3)} \end{bmatrix}$$

DEFINITION 3 (T-PRODUCT [21]). *: The t-product between two 3-order tensors with matched dimensions, $\mathcal{M} \in \mathbb{R}^{n_1 \times m \times v}$ and $\mathcal{N} \in \mathbb{R}^{m \times n_2 \times v}$, is defined as $\mathcal{M} * \mathcal{N} \in \mathbb{R}^{n_1 \times n_2 \times v}$, i.e.,*

$$\mathcal{M} * \mathcal{N} = \text{ifft}(\text{bdiag}(\overline{MN}))$$

*where $\overline{\mathcal{M}}$ is the discrete Fourier transform of $\mathcal{M}$ along the third dimension, $\overline{\mathcal{M}} = \text{fft}(\mathcal{M}, [], 3)$, $\overline{M} = \text{bdiag}(\overline{\mathcal{M}})$ and $\text{bdiag}(\cdot)$ denotes the block diagonal matrix.*

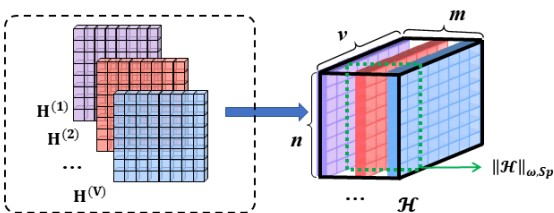

**Figure 1: Tensor Construction and Weighted Tensor Schatten $p$-norm**

## 3 THE PROPOSED METHOD

In this section, we first present the motivation for our proposed scheme. After that, we present the details of our proposed multi-view clustering scheme based on deep tensor factorization and the optimization method of the scheme. Finally, we summarize our scheme and give the time complexity analysis of the scheme. We show some key notations used in the scheme in Table 1 for easier presentation.

### 3.1 Objective

In recent years, tensor factorization has emerged as a powerful tool for multi-view data processing, since it can better explore

**Table 1: Notations**

| Notations | Description |
| --- | --- |
| $\mathbf{M}$ | Matrix with upper bold case letters |
| $\boldsymbol{\mathcal{M}}$ | Tensor with bold calligraphy letters |
| $\overline{\boldsymbol{\mathcal{M}}}$ | Discrete Fourier transform of $\boldsymbol{\mathcal{M}}$ along the third dimension |
| $\boldsymbol{\mathcal{M}}^{(v)}$ | $v$-th frontal slice of $\boldsymbol{\mathcal{M}}$ |
| $tr(M), M^T$ | Transpose and trace of matrix $M$ |
| $\|\boldsymbol{\mathcal{M}}\|_F$ | The F-norm of tensor $M$ |
| $\boldsymbol{\mathcal{S}} \in \mathbb{R}^{n \times m \times v}$ | Data tensor after anchor processing |
| $\boldsymbol{\mathcal{H}}_i \in \mathbb{R}^{n \times l_i \times v}$ | Indication tensor for layer $i$ |
| $\boldsymbol{\mathcal{W}}_1 \in \mathbb{R}^{l_1 \times m \times v}$ | Weight tensor for layer 1 |
| $\boldsymbol{\mathcal{W}}_i \in \mathbb{R}^{l_i \times l_{i-1} \times v}$ | Weight tensor for layer $i > 1$ |
| $\psi_i$ | Short for $\boldsymbol{\mathcal{W}}_{i-1} * \boldsymbol{\mathcal{W}}_{i-2} * \ldots * \boldsymbol{\mathcal{W}}_1$ |
| $\|\boldsymbol{\mathcal{X}}\|_{\omega,S_p}$ | Weighted tensor Schatten $p$-norm |
| $\boldsymbol{\mathcal{M}}, \boldsymbol{\mathcal{N}}$ | Auxiliary Variables |
| $\boldsymbol{\mathcal{Y}}_1, \boldsymbol{\mathcal{Y}}_2$ | Lagrange multipliers |

the relationship between different views and the spatial structure within the view, which provides the possibility to deal with the correlation between multiple views. Similar to NMF, the objective function of tensor factorization can be described as follows:

$$\min_{\mathcal{H},\mathcal{W}} \|\boldsymbol{\mathcal{X}} - \boldsymbol{\mathcal{H}} * \boldsymbol{\mathcal{W}}\|_F^2$$
$$\text{s.t.} \quad \boldsymbol{\mathcal{H}} \geqslant 0, \boldsymbol{\mathcal{H}}^T * \boldsymbol{\mathcal{H}} = \boldsymbol{\mathcal{I}} \tag{3}$$

where $\boldsymbol{\mathcal{X}} \in \mathbb{R}^{n \times d \times v}$ is the original data tensor obtained by combining the data matrices of the different views. $\boldsymbol{\mathcal{H}} \in \mathbb{R}^{n \times k \times v}$ is the cluster indicator tensor. $\boldsymbol{\mathcal{W}} \in \mathbb{R}^{k \times d \times v}$ is the cluster weight tensor. This fusion of data from different views into a tensor for factorization can naturally leverage the relationships between views [24]. Fig. 1 depicts the construction of tensor by taking $\boldsymbol{\mathcal{H}}$ as an exmple. However, despite tensor factorization's theoretical superiority, its ability to handle nonlinear structures and learn deep representations is still limited. Inspired by deep learning, this paper presents an innovative deep tensor factorization framework to overcome the above challenges. By incorporating the concepts of deep learning into the tensor factorization process, our approach is able to not only capture the multidimensional structure of data but also learn deep and nonlinear representations of data.

$$\min_{\mathcal{H}_l,\mathcal{W}_i} \|\boldsymbol{\mathcal{X}} - \boldsymbol{\mathcal{H}}_l * \boldsymbol{\mathcal{W}}_l * \boldsymbol{\mathcal{W}}_{l-1} * \ldots * \boldsymbol{\mathcal{W}}_1\|_F^2$$
$$\text{s.t.} \quad \boldsymbol{\mathcal{H}}_l \geqslant 0, \boldsymbol{\mathcal{H}}_l^T * \boldsymbol{\mathcal{H}}_l = \boldsymbol{\mathcal{I}} \tag{4}$$

The deep tensor factorization method enables us to more effectively process multi-view data by extracting the deep and potential correlations between different views. The following experiments section proves that our method can significantly improve the performance of multi-view clustering.

Considering that directly performing deep tensor factorization on the original data matrix incurs high computational complexity, we choose to construct a anchor graph for each view, and a anchor tensor $\boldsymbol{\mathcal{S}} \in \mathbb{R}^{n \times m \times v}$ constructed from the anchor graphs of different views to replace the original data tensor $\boldsymbol{\mathcal{X}} \in \mathbb{R}^{n \times d \times v}$

as the input to the tensor factorization, and $m$ is the number of anchors. Since the number of anchors is much smaller than the original data dimension (i.e., $m << d$), it can effectively reduce the computational overhead.

To further exploit the complementary information of different views, we further minimize the tensor schatten $p$ norm of the tensorial form of the cluster indicator matrix, and Eq. (4) becomes:

$$\min_{\mathcal{H}_l,\mathcal{W}_i} \|\boldsymbol{\mathcal{S}} - \boldsymbol{\mathcal{H}}_l * \boldsymbol{\mathcal{W}}_l * \boldsymbol{\mathcal{W}}_{l-1} * \ldots * \boldsymbol{\mathcal{W}}_1\|_F^2 + \lambda\|\boldsymbol{\mathcal{H}}_l\|_{\omega,Sp}$$
$$\text{s.t.} \quad \boldsymbol{\mathcal{H}}_l \geqslant 0, \boldsymbol{\mathcal{H}}_l^T * \boldsymbol{\mathcal{H}}_l = \boldsymbol{\mathcal{I}} \tag{5}$$

The second term of objective function is regularization term based on weighted tensor Schatten $p$-norm of $\boldsymbol{\mathcal{H}}_l$, and $\lambda$ is a balancing hyperparameter. The weighted tensor Schatten $p$-norm is defined as follows

DEFINITION 4 (WEIGHTED TENSOR SCHATTEN $p$-NORM [16]). *Given* $\boldsymbol{\mathcal{X}} \in \mathbb{R}^{n_1 \times n_2 \times n_3}$, *Weighted tensor Schatten $p$-norm of* $\boldsymbol{\mathcal{X}}$ *is defined as*

$$\|\boldsymbol{\mathcal{X}}\|_{\omega,S_p} = \left(\sum_{i=1}^{n_3} \left\|\overline{\boldsymbol{\mathcal{X}}}^{(i)}\right\|_{\omega,S_p}^p\right)^{\frac{1}{p}}$$
$$= \left(\sum_{i=1}^{n_3} \sum_{j=1}^{min(n_1,n_2)} \omega_j * \sigma_j\left(\overline{\boldsymbol{\mathcal{X}}}^{(i)}\right)^p\right)^{\frac{1}{p}}. \tag{6}$$

*where* $0 \leq p \leq 1$, $\sigma_j(\overline{\boldsymbol{\mathcal{X}}}^{(i)})$ *is the $j$-th singular value of* $\overline{\boldsymbol{\mathcal{X}}}^{(i)}$, $\omega_j$ *is the weight of* $\sigma_j(\overline{\boldsymbol{\mathcal{X}}}^{(i)})$.

It is worth noting that the aim of weighted tensor Schatten $p$-norm for the regularization is to avoid the problem of all singular values contributing equally in ordinary tensor Schatten $p$-norm, which provides better robustness to data noise or outliers. Based on [27], we update the weights by $\left(\omega_j^k\right)^{t+1} = \frac{1}{\left(\delta_j^p\left(\mathbf{X}_f^k\right)\right)^t + \varepsilon}$, where $t$ and $\varepsilon$ denote the number of iteration and a small constant.

## 3.2 Initialization

Before optimizing the objective equation, following the work on deep non-negative matrix factorization for multi-view clustering [42, 43], we first initialize layer after layer of views. We first get $\boldsymbol{\mathcal{W}}_1^{(v)}$ by $\boldsymbol{\mathcal{S}}^{(v)} = \boldsymbol{\mathcal{H}}_1^{(v)} * \boldsymbol{\mathcal{W}}_1^{(v)}$ and then factorize $\boldsymbol{\mathcal{H}}_1^{(v)} = \boldsymbol{\mathcal{H}}_2^{(v)} * \boldsymbol{\mathcal{W}}_2^{(v)}$ layer by layer up to $\boldsymbol{\mathcal{W}}_l$. Besides, We initialize $\boldsymbol{\mathcal{Y}}_1 = \boldsymbol{\mathcal{Y}}_2 = 0$ and $N$ as the unit matrix.

## 3.3 Optimization

We use the Augmented Lagrange Multiplier (ALM) to optimize the objective Eq. (5), To perform an alternating direction minimization strategy, we introduce two auxiliary tensor variables $\boldsymbol{\mathcal{H}}_l = \boldsymbol{\mathcal{N}}, \boldsymbol{\mathcal{H}}_l = \boldsymbol{\mathcal{M}}$ where $\boldsymbol{\mathcal{N}} \geqslant 0$ are used, and then, the objective function can be reformularized to the following unconstrained optimization problem:

$$\min \mathcal{L}(\mathcal{H}_l, \mathcal{W}_1, \mathcal{W}_2, \ldots, \mathcal{W}_l, \mathcal{N}, \mathcal{M})$$

$$= \min_{\mathcal{N} \geqslant 0, \mathcal{H}_l^{\mathrm{T}} \mathcal{H}_l = \mathcal{I}} \| \mathcal{S} - \mathcal{H}_l * \mathcal{W}_l * \mathcal{W}_{l-1} * \ldots * \mathcal{W}_1 \|_F^2 \quad (7)$$

$$+ \lambda \| \mathcal{M} \|_{\omega, Sp} + \frac{\mu}{2} \left\| \mathcal{H}_l - \mathcal{N} + \frac{\mathcal{Y}_1}{\mu} \right\|_F^2 + \frac{\rho}{2} \left\| \mathcal{H}_l - \mathcal{M} + \frac{\mathcal{Y}_2}{\rho} \right\|_F^2$$

where the tensors $\mathcal{Y}_1 \in \mathbb{R}^{n \times k \times v}$ and $\mathcal{Y}_2 \in \mathbb{R}^{n \times k \times v}$ are Lagrange multipliers and $\mu, \rho$ are the penalty parameters. The optimization process can therefore be separated into five steps:

$\mathcal{W}_i$-**subproblem**: By fixing $\mathcal{W}_1, \mathcal{W}_2, \ldots, \mathcal{W}_{i-1}, \mathcal{N}, \mathcal{H}_i, \mathcal{M}$, the optimization Eq.( 7) becomes:

$$\min \| \mathcal{S} - \mathcal{H}_i * \mathcal{W}_i * \psi_{i-1} \|_F^2 \quad (8)$$

where $\psi_{i-1} = \mathcal{W}_{i-1} * \mathcal{W}_{i-2} * \ldots * \mathcal{W}_1$. First, implement the discrete Fourier transform (DFT) along the third dimension, the equivalent representation of Eq.(8) in the frequency domain becomes:

$$\min \sum_{v=1}^{V} \| \overline{\mathcal{S}}^{(v)} - \overline{\mathcal{H}_i}^{(v)} \overline{\mathcal{W}_i}^{(v)} \overline{\psi_{i-1}}^{(v)} \|_F^2$$

where $\overline{\mathcal{W}} = \mathrm{ftt}(\mathcal{W}, [], 3)$, and the others in the same way. So Making the equation zero gets the solution of Eq.(8 )is:

$$\overline{\mathcal{W}_i}^{(v)} = \overline{\mathcal{H}_l}^{(v)\dagger} \overline{\mathcal{S}}^{(v)} \overline{\psi_{i-1}}^{(v)\dagger} \quad (9)$$

where $\overline{\mathcal{H}_l}^{(v)\dagger}$ denotes MP inverse of $\overline{\mathcal{H}_l}^{(v)}$ [10].

$\mathcal{H}_i$-**subproblem**: By fixing $\psi_i, \mathcal{N}, \mathcal{M}$, the optimization Eq.(7 )becomes:

$$\min \| \mathcal{S} - \mathcal{H}_i * \psi_i \|_F^2 \quad (10)$$

Same as $\mathcal{W}_i$. First, perform the DFT of the equivalent representation of Eq.(10) in the frequency domain becomes:

$$\min \sum_{v=1}^{V} \| \overline{\mathcal{S}}^{(v)} - \overline{\mathcal{H}_i}^{(v)} \overline{\psi_i}^{(v)} \|_F^2$$

According to [10] the update rule for $\overline{\mathcal{H}_i}^{(v)} (i < l)$ is

$$\overline{\mathcal{H}_i}^{(v)} = \overline{\mathcal{H}_i}^{(v)}$$

$$\odot \sqrt{ \frac{\left[ \overline{\psi_i}^{(v)} (\overline{\mathcal{S}}^{(v)})^T \right]^+ + \left[ \overline{\psi_i}^{(v)} (\overline{\psi_i}^{(v)})^T (\overline{\mathcal{H}_i}^{(v)})^T \right]^-}{\left[ \overline{\psi_i}^{(v)} (\overline{\mathcal{S}}^{(v)})^T \right]^- + \left[ \overline{\psi_i}^{(v)} (\overline{\psi_i}^{(v)})^T (\overline{\mathcal{H}_i}^{(v)})^T \right]^+} } \quad (11)$$

Where $[A]^+ = (|A| + A)/2$, $[A]^- = (|A| - A)/2$.

$\mathcal{H}_l$-**subproblem**: By fixing $\mathcal{N}, \psi_l, \mathcal{M}$ Eq.(7) becomes:

$$\min_{\mathcal{H}_l^{\mathrm{T}} \mathcal{H}_l = \mathcal{I}} \| \mathcal{S} - \mathcal{H}_l \psi_l \|_F^2 + \frac{\mu}{2} \left\| \mathcal{H}_l - \mathcal{N} + \frac{\mathcal{Y}_1}{\mu} \right\|_F^2 + \frac{\rho}{2} \left\| \mathcal{H}_l - \mathcal{M} + \frac{\mathcal{Y}_2}{\rho} \right\|_F^2 \quad (12)$$

Eq.(12) can be reduced to

$$\max_{\left(\overline{\mathcal{H}_l}^{(v)}\right)^{\mathrm{T}} \overline{\mathcal{H}_l}^{(v)} = \mathrm{I}} \mathrm{tr} \left( \left(\overline{\mathcal{H}_l}^{(v)}\right)^{\mathrm{T}} \overline{\mathcal{B}}^{(v)} \right) \quad (13)$$

where $\overline{\mathcal{B}}^{(v)} = 2\overline{\mathcal{S}}^{(v)} \left(\overline{\psi_l}^{(v)}\right)^{\mathrm{T}} + \mu \overline{\mathcal{Q}_1}^{(v)} + \rho \overline{\mathcal{Q}_2}^{(v)}$, $\overline{\mathcal{Q}_1}^{(v)} = \overline{\mathcal{N}}^{(v)} - \frac{\overline{\mathcal{Y}_1}^{(v)}}{\mu}$ and $\overline{\mathcal{Q}_2}^{(v)} = \overline{\mathcal{M}}^{(v)} - \frac{\overline{\mathcal{Y}_2}^{(v)}}{\rho}$

According to [24], the solution of Eq.(13 )is

$$\overline{\mathcal{H}_l}^{(v)} = \overline{\Lambda}^{(v)} \left(\overline{V}^{(v)}\right)^{\mathrm{T}} \quad (14)$$

where $\overline{\Lambda}^{(v)}$ and $\overline{V}^{(v)}$ can be obtained by SVD $\overline{\mathcal{B}}^{(v)} = \overline{\Lambda}^{(v)} \mathrm{X} \left(\overline{V}^{(v)}\right)^{\mathrm{T}}$

$\mathcal{N}$-**subproblem**: By fixing $\mathcal{H}, \mathcal{G}, \mathcal{M}$. Eq.(7) becomes:

$$\min_{\mathcal{N} \geqslant 0} \frac{\mu}{2} \left\| \mathcal{N} - \left( \mathcal{H} + \frac{\mathcal{Y}_1}{\mu} \right) \right\|_F^2 \quad (15)$$

According to [40], the solution of Eq.(15 ) is:

$$\mathcal{N} = \left( \mathcal{H} + \frac{\mathcal{Y}_1}{\mu} \right)_+ \quad (16)$$

$\mathcal{M}$-**subproblem**: By fixing $\mathcal{N}, \mathcal{H}, \mathcal{G}$. Eq.(7) becomes:

$$\min \lambda \| \mathcal{M} \|_{\omega, Sp} + \frac{\rho}{2} \left\| \mathcal{H} - \mathcal{M} + \frac{\mathcal{Y}_2}{\rho} \right\|_F^2$$

A simple transformation gives us

$$\mathcal{M}^* = \arg\min \frac{1}{2} \| \mathcal{H} + \mathcal{Z} \|_F^2 + \frac{\lambda}{\rho} \| \mathcal{M} \|_{\omega, Sp}, \quad (17)$$

where $\mathcal{Z} = \frac{\mathcal{Y}_2}{\rho} - \mathcal{M}$. According to [17], the following lemma can give the solution of Eq.(17)

LEMMA 1. *Let* $\mathcal{Z} \in \mathbb{R}^{n_1 \times n_2 \times n_3}$ *satisfy an increasing sequence of* $0 \leq \omega_1 \leq \omega_2 \leq \cdots \leq \omega_{\min}$ *and have a t-SVD* $\mathcal{Z} = \mathcal{U} * \mathcal{S} * \mathcal{V}^T$. *For the following optimization model:*

$$\underset{\mathcal{X}}{\arg\min} \frac{1}{2} \| \mathcal{X} - \mathcal{Z} \|_F^2 + \tau \| \mathcal{X} \|_{\omega, S_p}^p. \quad (18)$$

*a global optimal solution to the 18 is*

$$\mathcal{X}^* = \Upsilon_{\tau * \omega}(\mathcal{Z}) = \mathcal{U} * ifft \left( \mathrm{P}_{\tau * \omega}(\overline{\mathcal{Z}}) \right) * \mathcal{V}^T,$$

*where,* $\mathrm{P}_{\tau * \omega}(\overline{\mathcal{Z}})$ *is a tensor and* $\mathrm{P}_{\tau * \omega}\left(\overline{\mathcal{Z}}^{(i)}\right)$ *is the i-th frontal slice of* $\mathrm{P}_{\tau * \omega}(\overline{\mathcal{Z}}) = \mathrm{diag}(\gamma_1, \gamma_2, \ldots, \gamma_l)$ *and* $\gamma_i = T_p^{GST}\left(\sigma_i(\overline{\mathcal{Z}}), \tau * \omega_i\right)$, *the introduction of GST can be found in [17].*

Through ***Lemma***1, we get the solution of Eq.( 17) as

$$\mathcal{M}^* = \Upsilon_{\frac{\lambda}{\rho} * \omega} \left( \mathcal{H} + \frac{\mathcal{Y}_2}{\rho} \right) \quad (19)$$

The optimization of our multi-view clustering method based on deep tensor factorization can be summarized as Algorithm 1. It is worth noting that we need to optimize $\mathcal{H}_i$ and $\mathcal{M}_i$ layer by layer before obtaining $\mathcal{H}_l$.

## 3.4 Computational Complexity

To simplify the analysis, we set the dimension of each layer to be uniformly $d$ and assume that the number of iterations of the Initialization process and training process are $t_{pre}$ and $t_{train}$. The computational complexity of constructing the anchor data tensor from the original data tensor is $\mathcal{O}(vnmd + vnmlog(m))$. The computational complexity of is $\mathcal{O}(vt_{pre}nmd^2l)$ after approximate simplicity. The computational complexity of the update process for $\mathcal{W}_i, \mathcal{H}_i, \mathcal{H}_l, \mathcal{M}, \mathcal{N}$ are $\mathcal{O}(vlt_{train}(dnm+d^2m+d^2n))$, $\mathcal{O}(vlt_{train}d^2n + vlt_{train}dnm)$, $\mathcal{O}(t_{train}v(m^2k+mk^2))$, $\mathcal{O}(t_{train}vnk)$, $\mathcal{O}(t_{train}(2vnklog(vk) + v^2kn))$. As these parameters are small constants and $m < n, k < d$,

---

**Algorithm 1** Multi-view Clustering based on Deep Non-negative Tensor Factorization (MVC-DNTF)

---

**input:** multi-view data $\{\mathbf{X}^{(v)}\}_{v=1}^{V}$; anchor number $m$; cluster number $k$; number of layers and dimension for each layer;

**output:** Cluster labels $\mathbf{Y}$ of each sample.

1: **Initialize**: $\mathcal{H}_i^{(v)}, \mathcal{W}_i^{(v)}, \mu = 10^{-5}, \rho, \eta, \mathcal{Y}_1, \mathcal{Y}_2, \overline{\mathrm{N}}^{(v)}$ according to section 3.2.

2: Compute anchor graph matrix $\mathbf{S}^{(v)}$ of each view.

3: **while** not convergence **do**

4:    **for** $i < m$ **do**

5:       Update $\overline{\mathcal{W}_i}^{(v)}$ by solving Eq.(9)

6:       Update $\overline{\mathcal{H}_i}^{(v)}$ by solving Eq. (11).

7:    **end for**

8:    Update $\overline{\mathcal{W}_l}^{(v)}$ by solving Eq.(9).

9:    Update $\overline{\mathcal{H}_l}^{(v)}$ by solving Eq.(14).

10:   Update $\overline{\mathcal{N}^{(v)}}$ by solving Eq.(16).

11:   Update $\mathcal{M}$ by using Eq.(19).

12:   Update $\mathcal{Y}_1, \mathcal{Y}_2, \mu$ and $\rho$: $\mathcal{Y}_1 = \mathcal{Y}_1 + \mu(\mathcal{H} - \mathcal{N})$, $\mathcal{Y}_2 = \mathcal{Y}_2 + \mu(\mathcal{H} - \mathcal{M})$, $\mu = \min(\eta\mu, 10^{13})$, $\rho = \min(\eta\rho, 10^{13})$.

13: **end while**

14: Calculate the $K$ clusters by using $\mathbf{H} = \sum_{v=1}^{V} \mathcal{H}^{(v)}/V$.

15: **return** Clustering result (In the resulting indicator matrix, the index of the largest element in each row corresponds to the cluster label of the respective sample. ).

---

and in general $n$ is the largest number among them, the final computational complexity of our proposed scheme is $\mathcal{O}(vt_{pre}nmd^2l + vlt_{train}dnm + t_{train}vm^2k)$

# 4 EXPERIMENTS

## 4.1 Dataset

We evaluate the performance of the proposed method on eight widely adapted multi-view learning benchmark datasets, which are **3-sources**, **BBCSport** [18], **HW** [12], **Sonar** [32], **Yale**, **Vehicle Sensor** [11], **Caltech-5V** [14],**SentencesNYU v2(RGB-D)** [33]. Details of these datasets are shown in Table 2.

## 4.2 Compared Method

All experiments were conducted on a desktop computer equipped with a 13th Gen Intel(R) Core(TM) i5-13400 processor at 2.50 GHz, and 32 GB RAM. The experiments were executed using MATLAB 2023a (64-bit) as the primary software environment. For experimental setup, we have selected the following nine representative multi-view clustering algorithms to compare with our proposed method: Two multi-view clustering methods based on graph learning **GMC** [36] and **UDBGL** [13], Three sub-space multi-view learning methods **DiMSC** [5], **MvLRSSC** [2] and **RMSL** [26], a fast calculation method **FastMICE** [19] and two non-negative matrix factorization methods **MvDGNMF** [25] and **MVC-DMF-PA** [42], and a method for orthogonal non-negative tensor factorization **Orth-NTF** [24].

## 4.3 Experimental Setup

Before performing clustering, for all the methods including the proposed method and all the comparison methods, we first preprocess the data of all the datasets, i.e., we normalize the data of different views. For the method we proposed, $\lambda$ and $p$ are hyper-parameters. According to [42], when performing a two-layer tensor factorization the layer size is $[l_1, k]$,where $l_1$ is chosen from $[4k, 5k, 6k]$, when performing a three-layer tensor factorization the layer size is $[l_1, l_2, k]$, where $l_1$ is chosen from $[8k, 10k, 12k]$ and $l_2$ is chosen from $[4k, 5k, 6k]$ and so on. We used three widely used metrics to evaluate clustering performance i.e., 1) Accuracy (ACC); 2)Normalized Mutual Information(NMI); 3) Purity(PUR). For all the metrics mentioned previously, a higher value indicates better clustering outcomes. For each experiment to avoid the effect of random initialization, we repeated each experiment 10 times.

## 4.4 Experiment Results

The experimental results in Tables 3 and 4 demonstrate the performance of our method MVC-DNTF on eight datasets and are compared with several benchmark algorithms. The best results are bolded and the second best results are underlined. The main results of our analysis of these results are as follows:

(1) Our method performs better than the benchmark methods on most of the datasets. We believe that this superior performance is largely attributable to the unique design of our proposed method, which performs deep factorization based on a data tensor consisting of multi-view data anchor graphs. This design allows our method to effectively utilize spatial structure information and depth and hidden features from different views. Meanwhile the imposition of orthogonal and non-negative constraints enhances the interpretability of our clustering, and each row of the metrics matrix for each view is explicitly mapped to a cluster without the need for post-processing for label determination.

(2) Compared with DMVC and AwDMVC methods that rely on the deep semi-NMF framework, our method achieves superior results. This demonstrates that the tensor factorization method we adopt better preserves the structural information of the data in multiple dimensions and effectively captures the complementary information and intrinsic connections among different views. Compared to Orth-NTF, which also employs tensor factorization, our method also achieves superior performance, which suggests that our proposed deep tensor factorization can automatically learn deeper feature representations from multi-view data, and thus better capture the essential properties and intrinsic structure of the data.

In conclusion, the experimental results highlight the effectiveness of our multi-view clustering method based on deep tensor factorization.

## 4.5 Convergence

We performed convergence experiments on four different datasets. These experiments were aimed at observing the evolution of the clustering accuracy of the algorithm and the convergence behavior of the algorithm's objective function during successive iterations.

**Table 2: Datasets used in our experiments**

| Dataset | Type | Views number | View Dimension | Sample number | Cluster number |
|---|---|---|---|---|---|
| 3-sources | text | 3 | 3056/3631/3068 | 169 | 3 |
| BBCSport | text | 2 | 3283/3183 | 544 | 5 |
| HW | handwritten | 6 | 216/76/64/6/240/47 | 2000 | 10 |
| Sonar | signal | 3 | 20/20/20 | 208 | 2 |
| Yale | image | 2 | 1024/4096 | 165 | 11 |
| Vehicle Sensor | sensor | 4 | 5/5/7/5 | 1594 | 2 |
| Caltech-5V | image | 5 | 40/254/1984/512/928 | 1400 | 7 |
| SentencesNYU v2 (RGB-D) | image | 2 | 2048/300 | 1449 | 13 |

**Table 3: Clustering performance comparison in terms of ACC(%), NMI(%), and PUR(%) on 3-sources, BBCSport, HW, and Sonar datasets.**

| Datasets | 3-sources | | | BBCSport | | | HW | | | Sonar | | |
|---|---|---|---|---|---|---|---|---|---|---|---|---|
| Metrics | ACC | NMI | PUR | ACC | NMI | PUR | ACC | NMI | PUR | ACC | NMI | PUR |
| DiMSC | 69.23 | 63.13 | 74.56 | 85.85 | 70.62 | 85.85 | 24.50 | 12.19 | 25.85 | 55.77 | 1.25 | 55.77 |
| MvLRSSC | 55.92 | 49.81 | 70.59 | 62.87 | 40.47 | 64.63 | 66.88 | 68.88 | 70.56 | 50.48 | 3.12 | 53.37 |
| RMSL | 31.95 | 14.46 | 41.42 | 76.63 | 72.36 | 76.63 | 81.38 | 78.82 | 81.50 | 62.02 | 4.29 | 62.02 |
| GMC | 70.74 | 65.15 | 79.29 | 80.33 | 73.89 | 84.01 | 84.80 | 89.13 | 87.25 | 50.48 | 4.50 | 53.37 |
| MvDGNMF | 30.77 | 21.11 | 33.14 | 82.54 | 67.32 | 82.54 | 75.60 | 63.83 | 75.60 | 62.50 | 4.66 | 62.50 |
| UDGBL | 34.91 | 5.60 | 35.50 | 36.40 | 2.43 | 36.58 | 67.10 | 57.42 | 80.69 | 57.21 | 1.61 | 57.21 |
| FastMICE | 55.62 | 50.25 | 71.01 | 43.93 | 11.16 | 45.40 | 85.65 | 85.04 | 85.65 | 58.00 | 3.23 | 58.17 |
| MVC-DMF-PA | 53.84 | 20.02 | 56.21 | 73.34 | 52.68 | 76.28 | 86.90 | 76.58 | 86.90 | 53.84 | 53.27 | 53.84 |
| Orth-NTF | 72.78 | 72.78 | 77.50 | 89.15 | 79.49 | 89.52 | 89.35 | 86.16 | 89.35 | 97.11 | 83.80 | 97.11 |
| Ours | 83.43 | 76.66 | 88.16 | 94.85 | 87.85 | 94.85 | 97.60 | 95.24 | 97.60 | 97.11 | 83.80 | 97.11 |

**Table 4: Clustering performance comparison in terms of ACC(%), NMI(%), and PUR(%) on Yale, Vehicle Sensor, HAR, and RGB-D datasets.**

| Datasets | Yale | | | Vehicle Sensor | | | Caltech-5V | | | RGB-D | | |
|---|---|---|---|---|---|---|---|---|---|---|---|---|
| Metrics | ACC | NMI | PUR | ACC | NMI | PUR | ACC | NMI | PUR | ACC | NMI | PUR |
| DiMSC | 44.85 | 52.84 | 44.85 | 68.95 | 22.29 | 68.95 | 57.57 | 39.76 | 61.43 | 39.61 | 32.67 | 49.76 |
| MvLRSSC | 44.06 | 48.02 | 45.09 | 56.78 | 6.12 | 56.78 | 46.15 | 34.81 | 46.79 | 39.00 | 32.40 | 50.59 |
| RMSL | 78.78 | 78.23 | 79.39 | 67.50 | 11.90 | 67.50 | 55 | 52.18 | 59.07 | 12.63 | 2.85 | 26.98 |
| GMC | 21.21 | 27.51 | 24.24 | 80.43 | 28.75 | 80.43 | 34.07 | 48.4 | 36.07 | 40.23 | 33.06 | 46.51 |
| MvDGNMF | 36.36 | 42.70 | 38.79 | 50.06 | 0.60 | 50.06 | 49.57 | 38.24 | 53.86 | 26.50 | 0.78 | 27.26 |
| UDGBL | 34.91 | 5.60 | 35.50 | 36.40 | 2.43 | 36.58 | 31.8 | 23.54 | 19.28 | 57.21 | 1.61 | 57.21 |
| FastMICE | 62.42 | 57.01 | 65.46 | 51.49 | 0.085 | 51.69 | 77.58 | 69.6 | 79.57 | 41.81 | 32.61 | 49.53 |
| MVC-DMF-PA | 15.75 | 16.10 | 20.00 | 50.37 | 62.76 | 50.37 | 71.64 | 56.49 | 71.64 | 16.83 | 72.25 | 33.12 |
| Orth-NTF | 78.18 | 81.90 | 80.00 | 98.05 | 86.23 | 98.05 | 89.35 | 81.64 | 89.35 | 59.07 | 65.78 | 75.56 |
| Ours | 84.24 | 86.39 | 82.42 | 99.62 | 96.79 | 99.62 | 94.21 | 89.08 | 94.21 | 63.21 | 71.28 | 82.95 |

Fig 2 illustrates the convergence process of the algorithm on the four datasets. It is noteworthy that the algorithm converges rapidly after 60 iterations, with a significant decrease in the difference between the objective matrices $\mathcal{H}_l$, $\mathcal{M}$, and $\mathcal{N}$. At the same time, the clustering accuracy improves dramatically, which is consistent with the convergence of the objective values. This proves the effectiveness of our proposed method since the convergence of the objective function is equivalent to the improvement of the clustering performance.

## 4.6 Parameter Analysis

Our proposed algorithm depends on three key variable parameters: the number of anchors $m$, $\lambda$ and $p$. $\lambda$ and $p$ regulate the sparsity and orthology of the potential representation $\mathcal{H}_l$. The hyperparameter $\lambda$ plays an important role in the trade-off between the accuracy

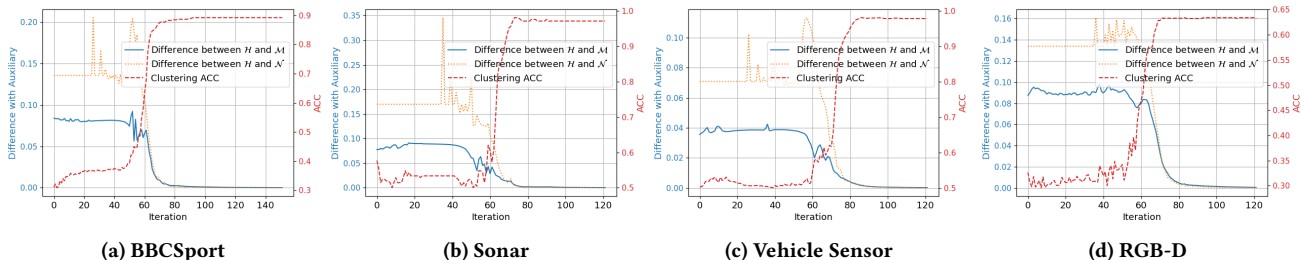

(a) BBCSport      (b) Sonar      (c) Vehicle Sensor      (d) RGB-D

Figure 2: Convergence experiments on BBCSport, RGB-D, Sonar and Vehicle Sensor

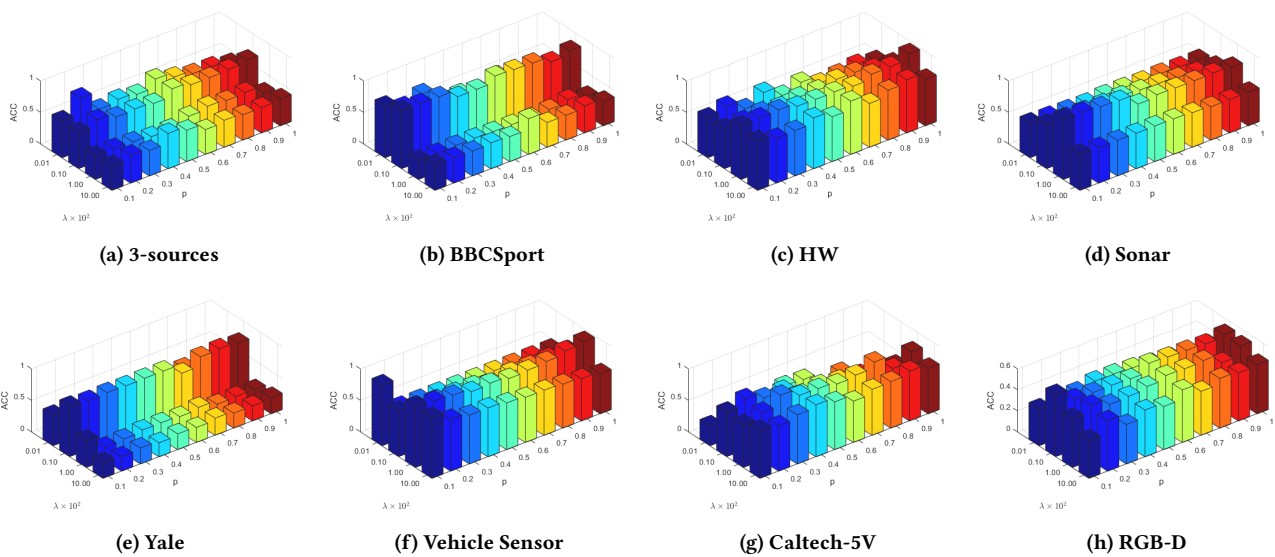

(a) 3-sources      (b) BBCSport      (c) HW      (d) Sonar

(e) Yale      (f) Vehicle Sensor      (g) Caltech-5V      (h) RGB-D

Figure 3: The influence of $\lambda$ and $p$ on clustering results on 3-sources, BBCSport, HW, Sonar, Yale , Vehicle Sensor Caltech-5V and RGB-D

of the data reconstruction and the sparsity of the representation. The hyperparameter $p$ fine-tunes this sparsity constraint by adjusting the distribution of non-zero values in $\mathcal{H}_l$, thus affecting the granularity of the representation. In order to study the impact of these hyperparameters in depth, we employ a grid search strategy to study values of $p$ in the range [0.1, 1.0] and values of $\lambda$ in the range [1, 1000]. We performed systematic experiments on eight different datasets to ensure the generalizability of our findings. Fig 3 illustrates the results of our experiments, which show that $\lambda$ has a significant effect on the efficacy of the algorithm, probably due to its key role in preventing overfitting through regularization, while $p$ has a relatively stable effect on performance, but tends to perform better when $p$ takes smaller values than when $p$ takes larger values. Empirical results from various datasets indicate that the best results are obtained when $\lambda$ is calibrated in the middle range 10 to 100 and $p$ is taken to be 0.1. This range allows the algorithm to achieve sufficient sparsity for generalization without compromising the ability to capture important data structures.

We set the anchor point rate from 0.1 to 1 with a step size of 0.1, and conduct experiments on the four datasets 3-sources , BBCSport,

Yale , RGB-D to test the effect of anchor point rate on the clustering results. As shown in Fig. 4, we can find that the clustering performance is improved with the increase of the number of anchors, but the anchor rate has little effect on the experimental results after the anchor rate is greater than 0.3. The four datasets achieve the best clustering results on 1.0, 0.9, 1.0, 0.8 respectively.

## 4.7 Ablation Study

We primarily focused on exploring how different depths of deep tensor factorization affect clustering performance for multi-view data. We executed numerous experiments using eight benchmark datasets as described in Section 4.1. These experiments varied the number of layers in the models to examine their influence on clustering results. The depths ranged from one to four layers, with configurations from $[k]$ to $[l_1, l_2, l_3, k]$. This setup allowed us to evaluate the effectiveness of deep tensor factorization at various layer depths. Results detailing the experimental accuracy rates are presented in Table 5.

**Table 5: ACC(%) of different layers on eight benchmark datasets.**

| $p$ | Caltech-5V | BBCSport | Yale | Vehicle Sensor | RGB-D | Sonar | HW | 3-sources |
|---|---|---|---|---|---|---|---|---|
| $[k]$ | 90.64 | 89.52 | 79.39 | 98.05 | 59.07 | 97.11 | 92.80 | 68.04 |
| $[l_2, k]$ | 90.64 | 89.15 | 80.00 | 98.05 | 61.07 | 97.11 | 97.60 | 76.33 |
| $[l_1, l_2, k]$ | 94.21 | 94.85 | 81.21 | 99.62 | 62.59 | 97.11 | 91.60 | 83.43 |
| $[l_1, l_2, l_3, k]$ | 89.57 | 90.07 | 84.24 | 98.05 | 63.21 | 97.11 | 96.35 | 75.73 |

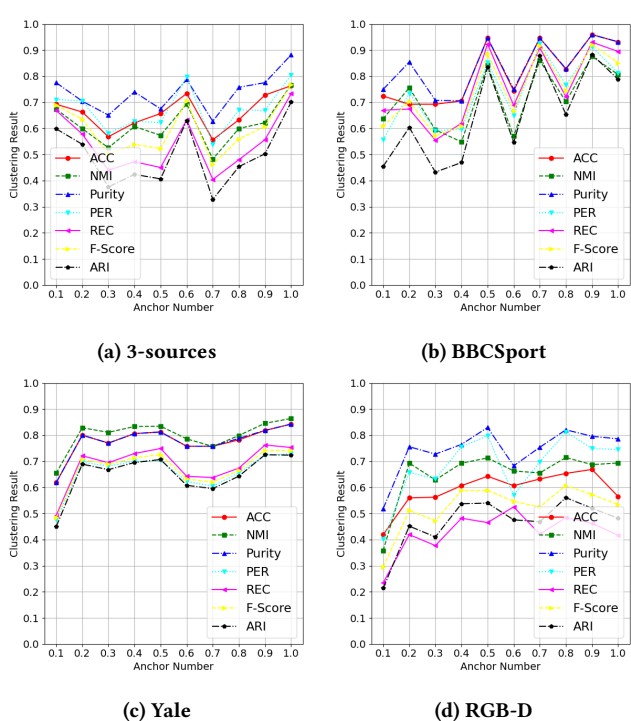

(a) 3-sources    (b) BBCSport

(c) Yale    (d) RGB-D

**Figure 4: Clustering results with different anchor rate on 3-sources, BBCSport, Yale, RGB-D**

On most datasets, the accuracy rate shows different trends as the layers deepens. For example, on the Caltech-5V , BBCSport , Vehicle Sensor 3-sources and HW dataset, the highest correctness rate was achieved using a three-layer tensor factorization ($[l_1, l_2, k]$), while four-layer tensor factorization ($[l_1, l_2, l_3, k]$) performed best on the Yale and RGB-D datasets.

However, overfitting may occur as the number of factorization layers increases . On the Caltech-5V dataset, the accuracy using two layers was essentially the same as when using only one layer , increasing to three layers the accuracy went up but then declined when going to four layers and the same happens on the BBCSport and HW datasets.

On the Vehicle Sensor and Sonar datasets, the performance improvement with increasing layers is not significant, which we attribute to the simpler structure of these datasets, where the accuracy of the basic tensor factorization is already close to 100, and thus it is difficult to achieve a large improvement in the deeper factorization.

In general, the deep tensor factorization method significantly improves the experimental results, which proves that our proposed multi-view clustering method with deep tensor factorization can effectively extract the deep and hidden features of the data.

Besides, We conducted ablation experiments on deep tensor factorization and weighted tensor Schatten $p$-norm in Table 6. case1 indicates whether deep tensor factorization or single-layer tensor factorization was used. due to the lack of consistency constraints of the tensor factorization methods alone, it is difficult to learn consistent cluster representation matrices, and the effect is poor and difficult to compare, so we set case2 as the use of the weighted tensor Schatten $p$-norm or the use of the ordinary Schatten $p$-norm.It can be found that the weighted Schatten $p$-norm is overall superior to the Schatten $p$-norm because the weighted Schatten $p$-norm capitalizes on certain important information corresponding to larger singular values while portraying complementary information from different views [27].

**Table 6: ACC(%) of ablation experiments**

| case1 | case2 | Datasets | | | |
|---|---|---|---|---|---|
| | | BBCSport | Yale | 3-sources | Caltech-5V |
| × | × | 89.15 | 78.18 | 72.78 | 89.35 |
| × | ✓ | 89.52 | 79.39 | 68.04 | 90.64 |
| ✓ | × | 91.91 | 78.18 | 81.06 | 90.00 |
| ✓ | ✓ | 94.85 | 84.24 | 83.43 | 94.21 |

## 5 CONCLUSION

We propose an novel multi-view clustering method based on deep tensor factorization named (MVC-DTF). By extending deem NMF to deep tensor factorization, MVC-DTF can effectively extract the deep and nonlinear features as well as the spatial informationof multi-view data . In addition, by introducing the weighted tensor Schatten $p$-norm of the clustering indicator matrix, the inter-view correlation and the complementary information of each views are well explored. We also develop an optimization algorithm for our proposed method. Extensive experiments and comparison with several state-of-the-art multi-view clusteirng method are conducted, whose results demonstrate the effectiveness and superiority of the method.

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
