# OpenReview forum: "Multi-view Clustering Based on Deep Non-negative Tensor Factorization"
_acmmm.org/ACMMM/2024/Conference — MM2024 Poster_

### Official Review · Reviewer_668h · 2024-05-21

**Rating:** 6
**Confidence:** 3

**Summary:**

The main idea of the work is multi-view clustering based on deep non-negative tensor factorization. The method stacks the matrix into a tensor and then decomposes it into the multiplication of several tensors, so as to extract the deep and nonlinear features of multi-view data. Weighted tensor schatten $p$-norm based regularization helps to exploit the spatial structure across different views. Additionally, anchor graph is introduced to reduce the computational complexity, and an optimization algorithm for deep non-negative tensor factorization is developed. The performance of the method is evaluated on several multi-view datasets, and the comparison with existing multi-view clustering algorithms demonstrate the superiority of the work.

**Strengths:**

The pros of the paper are:
1.The contributions seem to be sufficient, and the core idea (i.e., DNTF) is novel. The utilization of deep tensor factorization enables the extraction of deep and nonlinear features, which play a crucial role in attaining high-quality clustering results.
2.The extensive experiments on eight benchmark datasets are performed to evaluate the performance of the method. The results obtained from these experiments indicate the superior performance of the proposed method when compared to existing multi-view clustering approaches.
3.The ablation studies and parameter analyses demonstrate the robustness of the proposed method and the influence of different model components. These analyses provide a comprehensive evaluation of the method's performance under various conditions, which better supports the contributions of the paper.

**Limitations:**

The cons of the paper are:
1.The Introduction reviews most works related to multi-view clustering, but there are still some references to be further analyzed. The authors should add more related references.
2.Figure 2 and Figure 4 are somehow blur, and their quality could be further improved. Besides, the two figures are with different styles, and if possible, it is recommended to adjust them to follow a same style.
3.The paper is well-written with well-structured organization. However, there are still some typos. For example, the organization of the subsection “4.7 Ablation Study” might require adjustment; there are both MVC-DNTF and MVC-DTF in this paper, and the authors should use a same name.

**Suitability:**

3

---

### Official Review · Reviewer_TvUb · 2024-05-21

**Rating:** 5
**Confidence:** 3

**Summary:**

The paper introduces a multi-view clustering method based on deep non-negative tensor factorization (MVC-DNTF). The work aims to address the weakness of classical deep non-negative matrix factorization based multi-view clustering (MVC) method that they generally process each view independently and thus ignore the potential relationship between views. Therefore, the authors propose to extend the deep non-negative matrix factorization into deep non-negative tensor factorization (DNTF). To further explore complementary information between different views, a weighted tensor schatten-p norm based constraint is introduced. An optimization algorithm is developed to effectively solves the MVC-DNTF objective.  Experimental results on multiple benchmark datasets validate the proposed method's superior performance compared to existing multi-view clustering techniques.

**Strengths:**

(1) Tensorized methods have emerged as a popular way to process multi-view data in recent years. This work considers extending deep non-negative matrix factorization into deep non-negative tensor factorization, which is an innovative method for multi-view clustering tasks. The proposed deep non-negative tensor factorization helps to extract deep and nonlinear features from multi-view data and thus benefits the clustering performance.
(2) Apart from deep non-negative tensor factorization, a weighted tensor schatten-p norm based constraint is introduced to further improve the clustering results. This constraint can better exploit the complementary information between different views as well as the spatial structure.
(3) Deep non-negative tensor factorization seems to be more difficult to be solved. The work develops an effective optimization algorithm to solve the MVC-DNTF objective, which greatly strengthens the method's practical applicability.
(4) Extensive experiments conducted on eight benchmark datasets provide comprehensive validation of the proposed method's effectiveness and superiority over state-of-the-art algorithms.

**Limitations:**

1.	There are some typos and grammar mistakes in this paper to be corrected. For example, (1) there is a “tesnor” in page 1, which should be revised as “tensor”. (2) In table 1 (notations), \mathbfcal{\psi}_{i} should be \mathbfcal{\psi}_{i-1}.
2.	There are also some layout issues, especially in Section 3.4 when analyzing the computational complexity.
3.	Tensor construction could be further specified. The description of the tensor construction and the Schatten $p$-norm can be added in conjunction with Fig.1 for better presentation.

**Suitability:**

3

---

### Official Review · Reviewer_tgbR · 2024-05-22

**Rating:** 6
**Confidence:** 3

**Summary:**

This paper proposes a multi-view clustering method based on deep non-negative tensor factorization (MVC-DNTF). The approach aims to overcome limitations in existing non-negative matrix factorization methods by capturing nonlinear data structures and inter-view relationships. It introduces a weighted tensor Schatten p-norm regularization, which enhances the clustering results to effectively leverage the complementary information from different views. A deep non-negative tensor factorization method is developed to learn the deep information. The integration of the weighted tensor Schatten p-norm regularization and deep non-negative tensor factorization contributes to the improvement of the multi-view clustering performance. A comprehensive experiment is designed that evaluates the clustering performance of the work as well as the contribution of each component.

**Strengths:**

The paper is well-organized and presents an innovative idea of multi-view deep tensor factorization. Besides, a comprehensive experimental analysis is designed. In summary, its advantages include:

(1) The paper exhibits good organization and readability, with a well-structured presentation of the research. The literature review is relatively comprehensive, providing a sufficient overview of the relevant works in the field of multi-view clustering.

(2) The paper contributes by integrating deep non-negative Matrix Factorization concepts into non-negative tensor factorization, effectively capturing complex and nonlinear relationships within multi-view data. In this way, it effectively tackles limitations of conventional NMF-based methods, achieving improved clustering performance through deep feature extraction. Besides, the weighted tensor schatten-p norm works as a low-rank constraint to further enhance the clustering performance.

(3) The experimental results demonstrate the superiority of the proposed method over several state-of-the-art clustering algorithms across multiple datasets, thus validating its effectiveness for practical applications.

**Limitations:**

The paper confronts several limitations in writing and paper formatting, as summarized below:

(1) The writing would benefit from further improvement, particularly by simplifying lengthy sentences that can be difficult to understand. Additionally, there are a few typos that need to be addressed. In line 883 of Page 8, “case 1” should be corrected as “Case 1”.

(2) It is advisable for the authors to pay more attention to the formatting issue. Specifically, in Line 462, the formula extends beyond the boundary and requires adjustment to ensure proper display and readability.

**Suitability:**

3

---

### Official Review · Reviewer_kwfM · 2024-05-24

**Rating:** 5
**Confidence:** 4

**Summary:**

The paper presents a tensor-based multi-view clustering method termed MVC-DTF, which is derived from the non-negative matrix factorization based multi-view clustering (MVC-NMF). Nevertheless, it outperforms MVC-NMF since it can process multi-view data together within a tensor, rather than process each view independently. Thus it could better reveal the relationship between different views. Weighted tensor schatten-p norm and anchor graph are also employed to improve clustering performance and computational efficiency, respectively. An optimization to solve MVC-DTF is developed. The paper also includes a series of experimental and comparison results.

**Strengths:**

(1) Novelty: The idea of multi-view clustering based on deep non-negative tensor factorization is novel which employs deep tensor factorization to learn the deep and nonlinear features and simultaneously process all the views together in the form of tensor.
(2) Sufficient Contribution: Several contributions are made in this work to multi-view clustering. The utilization of deep non-negative tensor factorization is useful in promoting the clustering performance. Besides, the authors further introduce weighted tensor schatten-p norm based term to explore the complementary inter-view spatial information for better clustering performance, and take advantage of anchor for better efficiency.
(3) Technically soundness: The objective and optimization of the proposed MVC-DTF is obtained via a series of theoretical derivation.
(4) Experiments: Various experiments are illustrated in this paper, including clustering performance evaluation and comparison, parameter sensitivity test, and the ablation study, which evaluates the MVC-DTF from different views.

**Limitations:**

(1) Typos: There exist some typos in the paper. For example, some symbols in Table 1 seem not proper. The matrix M in Table 1 should be in bold, which should be corrected as \bf{M}. Besides, it is recommended to use a consistent term of MVC-DNTF or MVC-DTF throughout the paper.
(2) Method: the description of the method might benefit from providing more details. Some derivation is somehow simple, and some details should be added. For example, Eq. 12 (in line 397 of page 4) is reduced to Eq. 13, but the authors did not give the reason. Therefore, this part should be enriched to help readers to learn the paper better.

**Suitability:**

3

---

### Meta-Review · Area_Chair_88Bh · 2024-07-01

**Recommendation:** Accept (Poster)
**Confidence:** 5

**Metareview:**

In the paper, the authors proposed a Deep Non-negative Tensor Factorization technique for multi-view data clustering task. The paper is well-structured and the presented multi-view deep tensor factorization approach is interesting and innovative. The experiments are also sufficient and convincing. The paper also receives four positive recommendations (A, A, WA, WA). Therefore, based on the recommendations of reviewers and the novelty of the paper, the paper can be accepted and presented on ACM MM.